# SonicGuard Sensor—A Multichannel Acoustic Sensor for Long-Term Monitoring of Abdominal Sounds Examined through a Qualification Study

**DOI:** 10.3390/s24061843

**Published:** 2024-03-13

**Authors:** Zahra Mansour, Verena Uslar, Dirk Weyhe, Danilo Hollosi, Nils Strodthoff

**Affiliations:** 1Division AI4Health, Department for Health Services Research, Faculty of Medicine and Health Sciences, Carl von Ossietzky Universität Oldenburg, 26129 Oldenburg, Germany; zahra.mansour@uol.de; 2Fraunhofer IDMT, Institute Part HSA, 26129 Oldenburg, Germany; danilo.hollosi@idmt.fraunhofer.de; 3University Clinic for Visceral Surgery, Faculty of Medicine and Health Sciences, Carl von Ossietzky Universität Oldenburg, 26121 Oldenburg, Germany; verena.uslar@uol.de (V.U.); dirk.weyhe@pius-hospital.de (D.W.)

**Keywords:** stethoscope, biomedical signal, audio system, microphone, medical instruments, wearable sensors, wearable health monitoring, machine learning

## Abstract

Auscultation is a fundamental diagnostic technique that provides valuable diagnostic information about different parts of the body. With the increasing prevalence of digital stethoscopes and telehealth applications, there is a growing trend towards digitizing the capture of bodily sounds, thereby enabling subsequent analysis using machine learning algorithms. This study introduces the SonicGuard sensor, which is a multichannel acoustic sensor designed for long-term recordings of bodily sounds. We conducted a series of qualification tests, with a specific focus on bowel sounds ranging from controlled experimental environments to phantom measurements and real patient recordings. These tests demonstrate the effectiveness of the proposed sensor setup. The results show that the SonicGuard sensor is comparable to commercially available digital stethoscopes, which are considered the gold standard in the field. This development opens up possibilities for collecting and analyzing bodily sound datasets using machine learning techniques in the future.

## 1. Introduction

The sound generated by body organ execution encodes information about its physical condition. Accordingly, auscultation represents a common, valuable, noninvasive, low-cost, and efficient diagnostic tool [1].

Body sounds: Heart sounds originate from blood flowing through the heart chambers, as well as from the opening and closure of heart valves during the cardiac cycle [2]. Utilizing heart sounds has played an essential role for over 180 years in the early detection of cardiovascular irregularities, which can facilitate prompt treatment and help prevent subsequent cardiovascular complications [3]. Similarly, lung sounds carry valuable information about airflow within and around the lungs. The presence of abnormal breath sounds, such as crackles and wheezes, can indicate pulmonary illnesses [4]. Refs. [5,6] demonstrate the results of the movement of liquids and gases through the intestine during the digestion process. Monitoring bowel sounds often involves longer data acquisition periods. This is because gastrointestinal sounds do not occur continuously. For instance, an initially healthy individual might exhibit no bowel sounds for a period of up to 4 minutes, only to later demonstrate over 30 distinct sounds per minute during examination. Consequently, the importance of continuous monitoring becomes necessary to accurately assess bowel sounds [7]. Detecting differences in bowel sounds may lead to a better comprehension of bowel anatomy [8]. The use of bowel sounds has been constrained by the lack of efficient tools that enable prolonged auscultation to capture fluctuations in bowel sounds [9].

From analog stethoscopes to better diagnostic tools: While stethoscopes were initially introduced in the early nineteenth century, it took more than hundred years for a revolutionary leap in auscultation to occur with Dr. David Littmann’s patenting of diaphragm-based stethoscopes. This was followed by the development of digital stethoscopes in 1999, which are capable of recording and playing back body sounds [10]. The need for a new tool to monitor body sounds became even more crucial during the early stages of the COVID-19 pandemic. The combination of the high risk of infection and the intensified use of stethoscopes added to the challenges faced by doctors [10]. Moreover, the effectiveness of using a stethoscope heavily relies on the user’s practical knowledge and ability to diagnose based on sound, which often leads to poor interobserver reliability [11]. In addition to the listed difficulties, the large size and high cost of traditional digital stethoscopes make it impractical to utilize multiple devices simultaneously to record body sounds from various points on the body. This limitation can obscure certain pathological indicators that could be valuable for diagnosis. For example, simultaneously recording bowel sound signals from all abdominal quadrants using multiple sensors could provide important insights, but the current limitations of traditional stethoscopes hinder such an approach [12]. Given this context, the development of an innovative tool to record body sound signals becomes imperative. Such a tool should offer multiple channels for recording, maintain a low-cost profile, possess a compact size, ensure user-friendliness, facilitate remote and continuous monitoring, and have the capacity to function beyond the confines of healthcare facilities. Moreover, it should integrate seamlessly with machine learning models for diagnostic purposes.

Related research and applications: In the last two decades, several studies focused on developing wearable stethoscopes by using battery technology, low-power embedded processors, and noninvasive integrated sensors [13]. These sensors mostly targeted continuous cardiovascular and respiratory monitoring ranging from standard printed circuit boards (PCBs) [14] and soft wearable stethoscopes (SWSs) [15] to vests combining multisensors [16]. The sensors designed for bowel sound monitoring remained limited to stationary use cases such as monitoring intensive care patients after abdominal surgery to early predict postoperative ileus, where the sensors are connected to a small bedside computer [17,18].

Contributions: In this study, we introduce the SonicGuard multichannel sensor (Fraunhofer IDMT, Oldenburg, Germany), which is designed for the long-term recording of body sounds in general and bowel sounds in particular. The performance of this novel sensor was assessed through a series of tests comparing it to established benchmarks, including the gold standard Littmann^®^CORE digital stethoscope (3M™, St. Paul, MN, USA), henceforth referred to as Littmann, and a second commercially available digital stethoscope known as Thinklabs One (Thinklabs Medical LLC, Centennial, CO, USA), henceforth referred to as Thinklabs. These comparisons were conducted under various conditions, such as using a gelatin box designed to replicate the acoustic characteristics of the abdomen, an iSTAN Healthcare CAE (iSTAN) phantom, and real patients; see Figure 1 for a schematic overview of the study design. Across all these testing scenarios, the SonicGuard sensor showed high acoustic performance and consistent quality with respect to the gold standard stethoscopes.

## 2. Materials and Methods

### 2.1. Body Sound Audio Sensors

We start with a description of the three acoustic sensors that are considered in this study: Littmann, Thinklabs, and the SonicGuard sensor. A visual overview is presented in Figure 2, and essential specifications related to these sensors are summarized in Table 1. The data used in this study have been extracted using the sensors of interest by disabling all the postprocessing features implemented in the respective sensor software in order to perform an equitable comparison of the sensor hardware. Notably, the table highlights a substantial disparity in size and weight among the considered body sound audio sensors. Specifically, the SonicGuard sensor, which we propose in this study, exhibited a remarkable reduction of approximately 95.7% in weight compared to the commercially available sensors. Moreover, the reduction in diameter contribute to a more comfortable monitoring experience, particularly during long-time monitoring.

Littmann CORE digital stethoscope: It is one of the most commonly used stethoscopes [19] that stands out for its high acoustic performance and consistent quality. It incorporates tunable technology, thereby allowing the listener to perceive different frequency sounds contingent on the contact surface [20]. For our study, we chose the 3M Littmann CORE, which features a double-sided stainless steel chest piece, a single-lumen binaural, an aerospace alloy-anodized aluminum headset, and the CORE digital attachment. This digital attachment permits the stethoscope to connect to the EKO application via Bluetooth, thereby enabling the visualization, recording, and sharing of body sounds [21]. The analog Littmann stethoscope offers a frequency range of 20–20,000 Hz, whereas the digital version’s software applies filters beyond users’ control, thereby altering the frequency range based on the application—such as heart sounds (20–200 Hz) and lung sounds (100–1000 Hz). However, up until now, there is no frequency range specified for bowl sounds [21].

Thinklabs One digital stethoscope: This stethoscope features a patented Thinklabs electromagnetic diaphragm technology. This design relies on the substantial electric field that exists between the diaphragm and the conductive plate behind it [22]. The diaphragm is situated in front of the stethoscope’s body directly facing a conductive plate. When the diaphragm vibrates due to sound waves, the resulting vibrations are detected as variations in the gap between the diaphragm and the rear plate [22]. Unlike a traditional Bell/diaphragm selection for the filter, Thinklabs employs various software filters, which can be controlled by interacting with the primary display. In our study, we deactivated all the filters to ensure a more accurate comparison.

SonicGuard sensor: The proposed SonicGuard sensor is a prototype designed as part of a research project that aims to utilize machine learning modules to estimate digestive system activity through the analysis of bowel sounds. This sensor has been designed with a focus on simplicity and compactness, thereby ensuring comfort and lightness. It utilizes an Infineon analog MEMS microphone with low self-noise characteristics, thereby preserving the recording quality to high standards. The microphone is enclosed within a 3D-printed housing featuring rounded edges, which offers both a streamlined aesthetic and performance-enhancing capabilities. An IP57 water and dust resistance is used for reliable functionality across various environments.

The SonicGuard sensor weighs merely 10 g and occupies a volume less than that of a standard-sized ping pong ball (3.4 mm^3^). Its adaptability is further emphasized through the ability to connect it to any sound card using AES cables. To ensure accurate readings and minimize disturbances from sensor movements, the SonicGuard sensor is securely affixed to the skin using double-sided 3M 1522 medical tape.

Constructed from standard, cost-effective components and enclosed in a 3D-printed casing, the SonicGuard sensor facilitates a notable reduction in production expenses. This affordability enables the use of multichannel sensors for real-time monitoring across multiple body areas. A picture of SonicGuard sensors connected to a SonicGuard platform is presented in Appendix A.

### 2.2. Experimental Setup

The first phase of the qualification study focused on scrutinizing the specifications of the SonicGuard sensor and evaluating its ability to deliver reliable performance. Subsequently, in the second part of the examination, the primary objective was to impartially assess the performance of the sensors of interest under controlled conditions. This phase involved testing the sensor with simulation systems such as the gelatin box and the iSTAN phantom.

As the examination progressed, the final stage entailed a real-world scenario in which the performance of the sensors was evaluated in the presence of actual patients. This assessment also incorporated gathering and analyzing feedback from these patients to gain a comprehensive perspective. A schematic overall of the different parts of the qualification study is presented in Figure 1.

#### 2.2.1. Basic Qualification Tests

To assess the reliability of the SonicGuard sensor across various instances, a total of six SonicGuard sensors were individually tested. Each sensor was placed at a distance of 1 meter from the sound source within an isolated room with free field conditions. A standardized reference signal, encompassing the frequency range of 1–20 kHz, was then employed. The performance of the SonicGuard sensors was subsequently compared to the calibrated microphone (MIC 255-1) to gauge their accuracy.

#### 2.2.2. Test Using the Gelatin Box

Gelatin box: To conduct a comparative evaluation of the performance of the body acoustic sensors under controlled conditions with reproducible parameters, a gelatin box phantom was constructed. This phantom aimed to simulate the physiological circumstances that give rise to bowel sounds. Tissue-mimicking materials (TMMs) have been extensively utilized in biomedical research, particularly for photoacoustic measurements. These materials replicate the key acoustic properties of biological tissues by employing substances that closely resemble their acoustic characteristics [23]. The degree of similarity between TMMs and actual body tissues is determined by factors such as the speed of sound, attenuation coefficient, and acoustic impedance [24]. In this study, a TMM was utilized within a specially designed gelatin box phantom, which was created to simulate the production of bowel sounds; see Appendix B for details.

Experimental protocol: To replicate of the bowel sound signal, a reference signal, was derived from the average of five different bowel sound signals, which were boosted by 20 dB to encompass the entire range of potential bowel sounds. This reference signal was then played through the MIC 255 microphone positioned within the gelatin box. In this way, the entire system effectively imitated the characteristics of bowel sounds; see Figure 3. This module served as the foundation for measuring various parameters, including frequency response, self noise, and the impact of the presence of an air gap between the stethoscope surface and the gelatin box on the stethoscope performance.

#### 2.2.3. Tests Using the iSTAN Phantom

iSTAN phantom: The iSTAN CAE healthcare phantom, henceforth referred to as iSTAN phantom, offers a platform for the physical evaluation of various clinical signs, such as heart, breath, and bowel sounds, palpable pulses, chest excursion, airway patency, etc. These signs are dynamically modeled using mathematical algorithms based on human physiology and pharmacology [26]. The phantom has the capacity to replicate the physiological conditions of various subjects. For our testing, we selected a 33-year-old male with a weight of 70 kg. We employed this phantom to evaluate the performance of the three acoustic sensors. Initially, the phantom was used to simulate bowel sounds, followed by its utilization to replicate heart and lung sounds. The phantom provides three options for bowel sounds—normal, hyperactive, and hypoactive. Additionally, it enables the independent control of sounds produced from each quadrant: right upper quadrant (RUQ), left upper quadrant (LUQ), right lower quadrant (RLQ), and left lower quadrant (LLQ).

Experimental protocol: Various scenarios were executed to test different aspects of the three sensors. Firstly, in a scenario where all four quadrants were active, each sensor was placed simultaneously on an active quadrant to record the same signal. The power spectrum density (PSD) was then computed for each sensor to determine which sensor could gather more information. In the next scenario, one quadrant was active while the other three were inactive. The three sensors were simultaneously placed on each of the inactive quadrants to evaluate their ability to capture signals from different directions. Lastly, in a separate test, the three sensors were positioned on the 2nd intercostal space (ICS) aortic area to record the normal S1–S2 heart sound. For lung sounds, the sensors were placed above the right clavicle to capture normal, crackle, and wheeze sounds.

Machine learning analysis: The particular experimental setup where the sensors were used to capture record identical signals can be exploited by using the recorded sounds as input for machine learning models. The goal was to assess the suitability of the different sensor data for downstream analysis using machine learning. Here, we created a dataset to distinguish three conditions of bowel sounds supported by the phantom: normal, hypoactive, and hyperactive. Three datasets have been created from the three sensors. Each record was sliced into records of six seconds in length, which resulted a dataset containing 411 records for each sensor corresponding to 137 samples per class. Subsequently, each dataset was split into a test set of 30% and a training set of 70%.

#### 2.2.4. Test with Patients

Subject information: The research protocol was approved by the Medical Ethics Committee of the University of Oldenburg (2022-056). The participants were informed about the purpose of the study and consented to the use of the collected data in an anonymized form. The performance of the acoustic sensors was assessed using healthy subjects from the medical field with no history of digestive system diseases. These subjects were sourced from the Pius Hospital in Oldenburg, Germany. The first evaluation involved a subgroup of 10 subjects (5 male and 5 females, age of 28.6±5.5 years, last meal at 189.5±198.5 min). The aim was to assess the acoustic sensors’ performance by concurrently placing the three sensors on the same quadrant of the subjects’ abdomen. The sensors recorded for 7 minutes, thus ensuring that they operated under identical conditions while capturing the same signal.

Sensor usability: A second subgroup consisting of 10 different subjects (5 male and 5 females, age of 31.66±7.4 years, last meal at 71.98±118.39 min) was used to evaluate the usability of the stethoscopes. Each stethoscope was used separately to examine a subject for two minutes. Subsequently, the subjects completed a survey assessing their user experience during the measurement. Further details about the survey can be found in Appendix D.

## 3. Results and Discussion

### 3.1. Basic Qualification Tests

The reliability of the SonicGuard sensor was evaluated through a comparison of delay [ms], root mean square error (RMS) [dB], and frequency response across six identical SonicGuard sensors in comparison to a standard reference microphone. The corresponding frequency responses are shown in Figure 4.

The mean and standard deviation of the delay for the SonicGuard sensors in comparison to the standard microphone were calculated as 0.014±0.003 ms and 4.617±0.926 mm, respectively. Additionally, the root mean squared error (RMS) was found to be 42.117±5.035 dB.

The differences in the frequency responses among the SonicGuard sensors were less than 2 dB, which is considered negligible and indicative of a high reliability. Furthermore, the SonicGuard sensor demonstrated the ability to replicate similar frequency responses with matching peaks and troughs to the calibrated microphone (Mic 255-1), thus exhibiting an amplitude shift of less than 20 dB.

### 3.2. Test Using the Gelatin Box

Frequency response: The frequency response of the body acoustic sensors was investigated using a function generator to produce a known test signal in the form of a sinusoidal waveform. This test signal encompassed a wide range of frequencies, starting from a low frequency of 20 Hz and incrementally increasing up to the upper limit of 20 kHz. The outcomes of this test are visually depicted in Figure 5 (top row). It shows that all acoustic sensors exhibit comparable performance within the frequency range of interest (100–1000 Hz), with amplitudes ranging between 80 to 100 dB. This signifies that all the sensors are proficient in detecting and transmitting signals across a similar frequency range.

Self-noise: A pivotal metric that characterizes the acoustic sensors’ performance is the self-noise. This parameter was assessed by recording the sensor output in a quiet room when no signal was present. The results, again shown in Figure 5 (top row), reveal that the self-noise for all sensors was markedly lower than the noise floor, which is the lower limit of the frequency response curve. This finding indicates that the sensors’ self-noise levels were comparatively small in comparison to the ambient background noise level, particularly within the desired frequency range of 1–1000 kHz.

Resilience to air gaps: Furthermore, a significant scenario explored during the gelatin box testing involved evaluating the acoustic sensor’s performance when it was partially attached to the abdominal wall—an occurrence frequently encountered in healthcare daily activity. A comparison of the acoustic sensor’s frequency response with and without a 2 mm air gap is depicted in Figure 5 (bottom row). It is evident that the presence of an air gap significantly degraded the acoustic sensor’s performance. In the case of the Littmann and SonicGuard sensors, the reduction in frequency response was limited to a maximum of 20 dB. However, the Thinklabs stethoscope experienced a substantial drop exceeding 40 dB, and in certain instances, the frequency response even aligned with the self-noise level. These findings underline the greater resilience of the Littmann and SonicGuard sensors, thus enabling them to maintain satisfactory performance even under slightly adverse conditions.

### 3.3. Tests Using the iSTAN Phantom

The iSTAN phantom presents a distinctive advantage in its ability to manipulate each quadrant as a bowel sound source, which is a capability that is not feasible with other simulation techniques or real patient scenarios. To assess the acoustic signal’s performance under various conditions, the power spectrum density (PSD) of the recorded signal was computed within the frequency range of interest [100–1000 Hz]. The PSD was chosen as the evaluation metric due to its capacity to offer insights into the frequency composition and energy distribution of the bowel sound signal.

Sensor performance during sound collection from a single direction: The objective of the initial test was to determine which acoustic sensor was most capable of recording optimal sound quality when positioned directly above the active quadrant, thereby encompassing varied conditions, such as normal, hyperactive, and hypoactive. The outcomes, detailed in Table 2, demonstrate that the SonicGuard and Littmann garnered the highest power spectrum density values when recording sound across diverse conditions. In particular, the SonicGuard exhibited the highest PSD in 86.67% of the cases.

Sensor performance during sound collection from different directions: When the acoustic sensor was positioned away from the direct location of the bowel sound (BS) source, the SonicGuard sensor consistently exhibited superior power spectrum density (PSD) compared to its counterparts, as shown in Table 2. This phenomenon can be attributed to the fundamental physical principles governing sound collection, which are due to diaphragm vibrations in the case of SonicGuard and air vibrations for MEMS microphones, which lack direct contact with the body. The SonicGuard’s heightened directional sensitivity potentially brings valuable advantages during auscultation, thereby representing a noteworthy advancement over existing tools.

Localization properties (Crosstalk test): This assessment focuses on exploring the advantages of multichannel recordings, which are a distinctive feature of the SonicGuard sensor. The aim of this test was to investigate the potential benefits of using multiple channels to precisely locate activity within specific quadrants of the abdomen. The power spectrum density (PSD) of the recorded signals, as detailed in Table 3, confirms the hypothesis that the SonicGuard sensor, when positioned directly above the active quadrant, captures the bowel sound signal with the highest PSD. This innovative capability holds significant potential in offering healthcare professionals a more comprehensive understanding of quadrant activity and facilitates meaningful comparisons between them, thereby potentially aiding in the early detection of abnormalities.

### 3.4. Analyzing Audio Sensor Data Using Machine Learning

Until now, we have evaluated the three sensors using several simulation modules in a descriptive fashion by means of parameters that evaluate the sound quality. However, the relevance of these parameters for potential downstream tasks operating on this data remains unclear. As a preliminary exploration of the potential of machine learning methods for analyzing body sounds and classifying different diseases, we conducted an assessment where we tried to infer three levels of bowel activity from the sensor recordings. Similar to any supervised machine learning module, our approach followed a structured framework, as illustrated in Figure 6:

Data collection: To construct a dataset for this experiment, each of the three sensors of interest was placed on the abdominal wall of the iSTAN phantom. We concurrently recorded bowel activity from all four quadrants. This dataset contains records of fixed duration (1 min) and presents a balanced representation of the three types of bowel sounds: normal, hypoactive, and hyperactive.

Feature extraction: In the feature extraction phase, we considered multiple feature sets of varying sizes. The first feature set encompassed three features covering the main three types of the fractal dimension [27]. Additionally, we explored the Geneva Minimalistic Acoustic Parameter Set (GeMAPS v01a) [28] comprising 62 commonly used audio features that assess signal characteristics in temporal, frequency, spectral, and energy domains. The largest feature set, with a total of 6373 features, is the ComParE 2016 [28] feature set, thereby encompassing a comprehensive set of audio features.

Classifier training and evaluation: In the classification phase, our goal was to construct, train, and test a classifier using the previously mentioned feature sets. We assessed the performance of each model by measuring its accuracy and studied how changing the sensor affected the classification process. To achieve this, we used a support vector machine (SVM) [29]) classifier. We used a hold-out set for performance evaluation, thus splitting the data into 70% for training and 30% for testing.

Our findings revealed that the SonicGuard sensor consistently exhibited the highest classification accuracy when compared to data recorded using the Littmann and Thinklabs sensors. This observation held true across all feature sets, with the SonicGuard reaching the highest performance at 92.7% accuracy, followed by 88.7% and 87.9% for Littmann and Thinklabs, respectively; see Figure 6 for a graphical representation of the results. These findings are largely consistent with those obtained by using different classifiers such as k-nearest neighbors or decision tree classifiers; see Appendix C for details. These were, however, outperformed by the SVM classifier.

Summary: This test suggests that choosing a high-quality sensor for recording bowel activity can significantly enhance the performance of machine learning models used for downstream analysis. Furthermore, it opens the door for future studies to test this hypothesis with more advanced machine learning techniques. In summary, these results underscore the value of utilizing phantom data for sensor qualification studies. Such data enable the investigation of identical signals with a well-defined ground truth through various sensors, thereby providing valuable insights into their performance.

### 3.5. Applicability to Heart and Lung Sounds

The SonicGuard sensor has been designed as a sensor for multichannel, long-term recordings of abdominal sounds. Similar to the competing stethoscopes considered in this study, the SonicGuard sensor can also be applied to capture different bodily sounds such as heart or lung sounds. A detailed qualification study in these domains is beyond the scope of this work. Nevertheless, we aim to demonstrate the feasibility of using SonicGuard in these application domains through additional experiments with the iSTAN phantom.

We acquired recordings of a healthy heart by individually placing each sensor on the second right intercostal space—a recognized standard point for capturing valve sounds [30]. The heart sounds from two beats are depicted in Figure 7. Similarly, the iSTAN phantom simulation generated authentic normal lung sounds when the sensor was positioned on the upper lobe [31], as illustrated in Figure 8. Importantly, all three acoustic sensors exhibited remarkable proficiency in capturing both heart and lung sounds. These data hold the potential to establish precise heart and respiratory cycles based on the collected information.

### 3.6. Test with Patients

Despite the accuracy of simulation systems, the validation of sensors using real patient data within clinical settings holds paramount importance. The initial patient test aimed to compare the performance of the acoustic sensors when recording the same signal simultaneously from the same subject.

Bowel activity in time and frequency domain: Bowel activity was manually annotated and analyzed both in the time and frequency domains for all 10 subjects. Illustrations for randomly selected bowel activity from random subject are provided in Figure 9. In the time domain representation, normal bowel activity is often characterized by high-pitched, gurgling, or tinkling sounds. These sounds exhibit a rhythmic pattern and can exhibit variations in frequency and intensity depending on the individual’s digestive activity. The three acoustic sensors under scrutiny are capable of detecting this activity with varying degrees of fidelity, thus capturing signal nuances.

The spectrum of bowel activity showcased a presence of white noise coupled with bowel activity recorded using the Thinklabs sensor in Figure 9. This phenomenon could stem from the deactivation of filters associated with the Thinklabs sensor. On the contrary, Littmann’s filters were designed internally and lacked provisions for user adjustments.

Time domain parameters: To achieve a comprehensive understanding of the acoustic sensor performance, various parameters were computed. The dynamic range, a measure of sound intensity and audio quality, demonstrated that the Littmann exhibited the best performance across all subjects (1.30±0.65), followed by the SonicGuard (0.70±0.35) and the Thinklabs (0.23±0.15). Zero-crossing values, which indicate instances of waveform crossings across the zero axis, revealed a substantial impact of white noise recorded by Thinklabs, thus yielding a notably high value (1803.4±528.3). This was followed by SonicGuard (241.65±79.5), with Littmann closely trailing (229.5±41.1). Similarly, the fractal dimension values, reflecting signal complexity and self-similarity, displayed Thinklabs with the highest value (1.63±0.23), thus succeeded by SonicGuard (1.33±0.12) and Littmann (1.07±0.05).

Time domain parameters after band-pass filtering: At this point, it is worth stressing that the Littmann did not allow access to the unfiltered raw signal but instead only provided the signal after the application of an unspecified band-pass filter. As the filtering process is expected to have a significant impact on the signal parameters, we repeated the analysis from above after applying a 6th-order Butterworth band-pass filter [20–1000 Hz] to the data recorded by the three sensors.

For dynamic range, the sensor performance remained consistent before and after filtration. Littmann maintained the highest value (1.9±0.98), followed by SonicGuard (0.58±0.28) and Thinklabs (0.32±0.28). Notably, the filtration had a substantial impact on the zero-crossing values, particularly for Thinklabs, which saw its mean value decrease from (1803.4±528.3) to (202.66±51.44). In contrast, Littmann (187.96±38.28) and SonicGuard (143.38±55.80) showed a lesser effect. The same trend was evident in fractal dimension values; Thinklabs underwent the most substantial reduction from (1.63±0.23) to (1.03±0.01). The mean and standard deviation values for these parameters were calculated across the ten subjects, both before and after filtration, and are presented in Table 4.

Usability assessment: The second experiment with patients focused on the user experience with the acoustic sensors. The responses of the subjects to the survey questions have been visually summarized in Figure 10. As the subjects were individuals working in the medical field, they were already familiar with the Littmann stethoscope, which had earned their trust over time. Simultaneously, they were intrigued by the small size of the SonicGuard sensor.

In response to the survey, 40% of the subjects favored the Littmann stethoscope, while an equal percentage of 40% expressed a preference for the SonicGuard sensor as the most comfortable acoustic sensor. A distinct subset of 20% indicated their comfort with both sensors. Similarly, when asked about the level of movement restriction, the proportions remained consistent.

Regarding extended recording sessions, the majority, accounting for 63%, favored the SonicGuard sensor. Notably, 20% of the participants reported experiencing thermally induced discomfort due to the cold surface of the Thinklabs diaphragm. On a related note, 20% of the respondents found that the SonicGuard sensor had the surface temperature most similar to their skin.

Summary: The different experiments carried out as part of the qualification study revealed that the SonicGuard sensor performs at a level comparable to the Littmann stethoscope, which is a commercially available stethoscope widely consider as the gold standard in the field. At the same time, the sensor has unique features such as multichannels, continuous monitoring enabled by its light weight, small size, and low cost. The mentioned specifications have a revelatory impact on bowel sound recording and promote its value for medical applications. In particular, the strong performance of the SonicGuard sensor for downstream tasks leveraging machine learning are a very promising sign for a forthcoming study of bowel sound activity using the SonicGuard sensor.

## 4. Conclusions

Using body sounds as a diagnostic tool is as ancient as the medicine itself; still, analogue stethoscopes are the most widely used technology for auscultation. Digital stethoscopes combined with modern data analysis techniques could represent an advantage in terms of diagnostic capabilities. In this work, we described the SonicGuard sensor, a novel, multichannel acoustic sensor for applications in long-term body sound monitoring. We thoroughly tested the sensor in different qualification tests with a particular focus on abdominal sounds and found it to perform on par with existing gold standard digital stethoscopes. We envision the sensor to have significant potential for future investigations of abdominal, lung, and heart sounds, which will be explored in the near future. By integrating the SonicGuard sensor with user-friendly mobile hardware, accompanied by an intuitive interface and enhanced by a machine learning module, the possibility emerges for a highly promising system. This system has the potential to effectively detect anomalies in bowel sounds, thus serving as a valuable tool for the early detection of diseases or the close monitoring of patients, for example after surgical procedures.

## Figures and Tables

**Figure 1 sensors-24-01843-f001:**
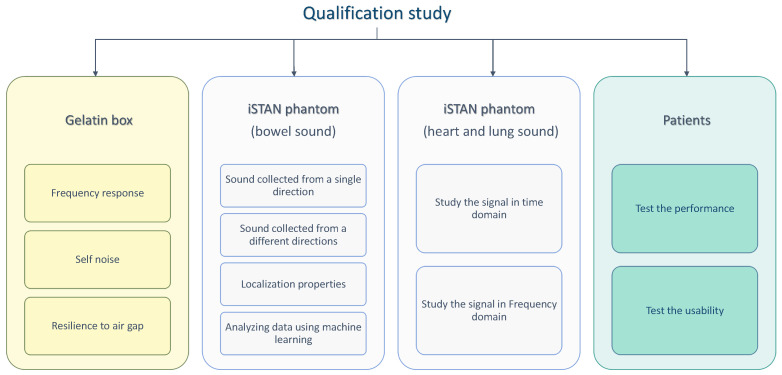
Schematic representation of the different qualification tests carried out as part of this study. The orange box symbolizes the fundamental qualification test using a gelatin box. The blue boxes denote tests utilizing iSTAN phantom. Lastly, the green boxes signify tests conducted on patients.

**Figure 2 sensors-24-01843-f002:**
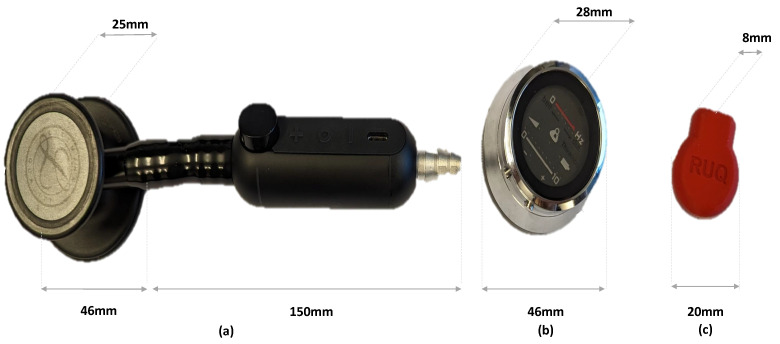
Sensors considered in this work: (**a**) commercial 3M Littmann Digital CORE, (**b**) commercial Thinklabs One Digital Stethoscope, and (**c**) proposed SonicGuard sensor.

**Figure 3 sensors-24-01843-f003:**
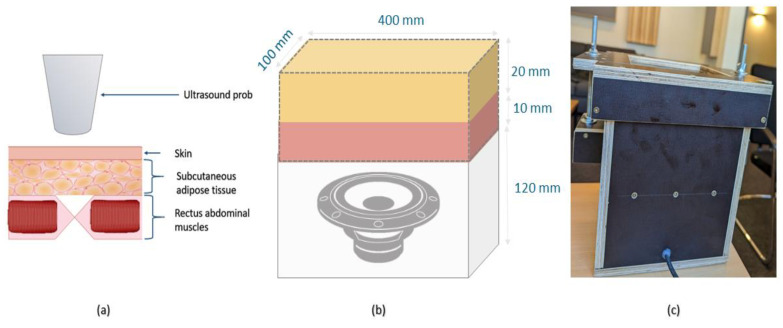
The gelatin box used to simulate the acoustic characteristic of the main abdominal layers. (**a**) The main anatomical structure of the abdominal layers comprises the subcutaneous adipose tissue, followed by the rectus abdominal muscles [25]. (**b**) The gelatin box’s 3D design and dimensions are as follows: The oil mixture gelatin phantom, mimicking subcutaneous adipose tissue, is represented in yellow, while the gelatin phantom imitating the rectus abdominal muscles is depicted in red. (**c**) A photograph of the utilized gelatin box for conducting the experiments is provided.

**Figure 4 sensors-24-01843-f004:**
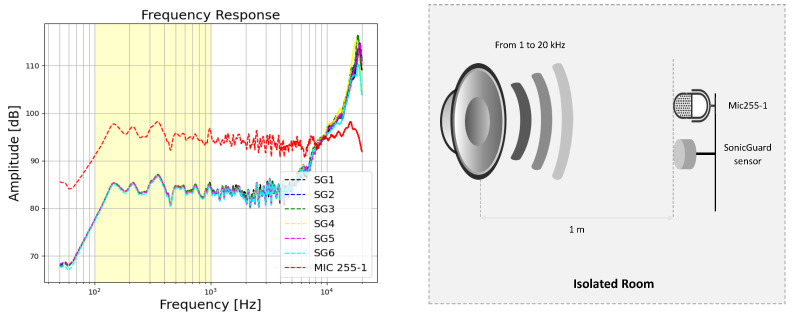
Left: The reliability of the SonicGuard sensor was evaluated by examining the absolute frequency response of all six instances of the SonicGuard sensor [SG1, SG2, SG3, SG4, SG5, SG6] in comparison to a reference microphone: Mic 255-1. The frequency range of interest (100–1000 Hz) is highlighted in the yellow box. This figure shows high similarity in frequency responses between the calibrated microphone (in red) and the 6 SonicGuard sensors, while the differences between the 6 SonicGuard sensors were negligible. Right: Experimental setup for the measurement represented by fixing the SonicGuard sensor and the calibrated microphone 1 m away from the sound source inside an isolated room.

**Figure 5 sensors-24-01843-f005:**
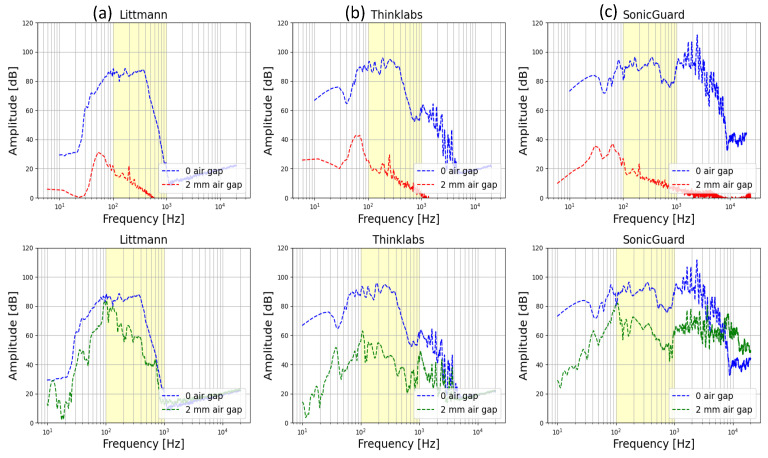
Top row: The frequency response (FR) in blue and the self-noise in red are shown for the three examined sensors: (**a**) Littmann, (**b**) Thinklabs, and (**c**) SonicGuard. This figure underscores that the self-noise levels for all three sensors were situated beneath the background noise. Bottom row: The frequency response of the examined sensors is shown in the case where the sensor was fully connected to the skin (blue) in comparison to the case where the sensors were displaced 2 mm from the examined surface (green). The figure demonstrates differences between the three sensors in their ability to perform reliably under different circumstances, where the Littmann and SonicGuard turned out to be the most resilient.

**Figure 6 sensors-24-01843-f006:**
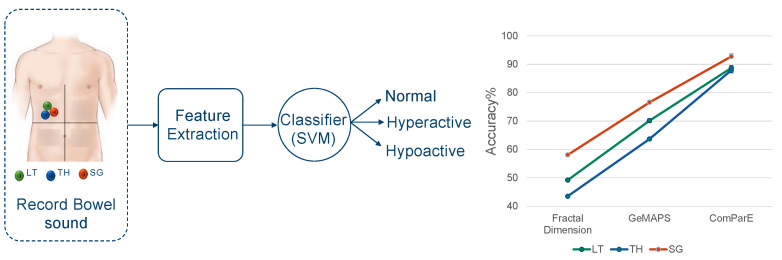
Left panel: Schematic representation of the machine learning approach, starting with bowel sound signal capture using three audio sensors (LT: Littmann, TH: Thinklabs, SG: SonicGuard). Signals are then transformed into a (tabular) feature representation and classified into normal, hyperactive, or hypoactive by means of an SVM classifier. Right panel: Accuracy of each sensor (green: Littmann, blue: Thinklabs, red: SonicGuard) across the three different feature sets (Fractal Dimension, ComParE2016, GeMAPS v01a).

**Figure 7 sensors-24-01843-f007:**
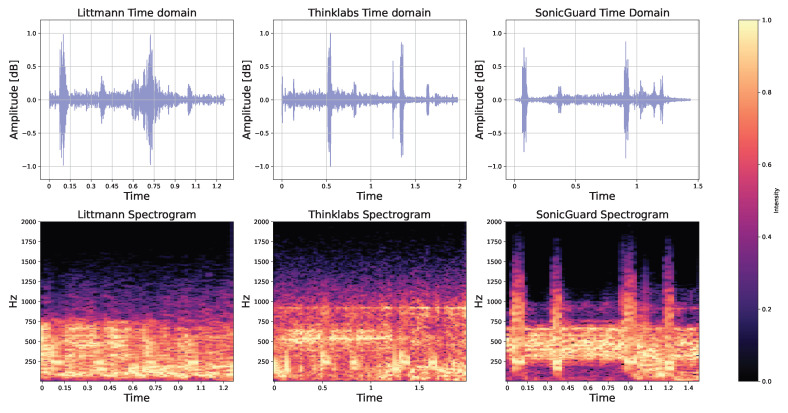
Normal heart sound signal from the iSTAN phantom recorded using Littmann, Thinklabs, and SonicGuard sensors in the time and frequency domain.

**Figure 8 sensors-24-01843-f008:**
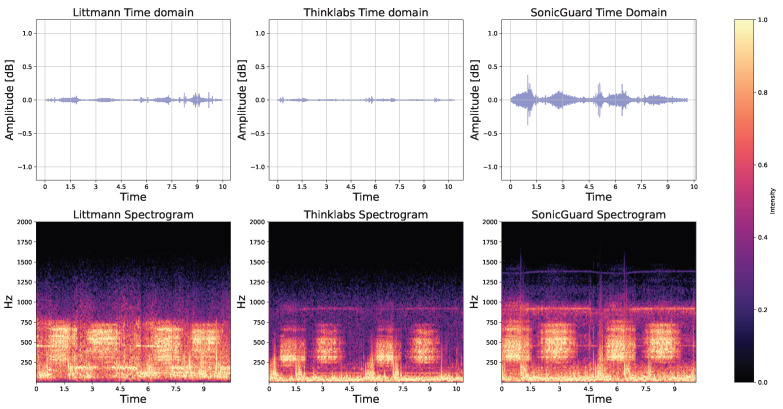
Normal lung sound signal from the iSTAN phantom recorded using Littmann, Thinklabs, and SonicGuard sensor in the time and frequency domain.

**Figure 9 sensors-24-01843-f009:**
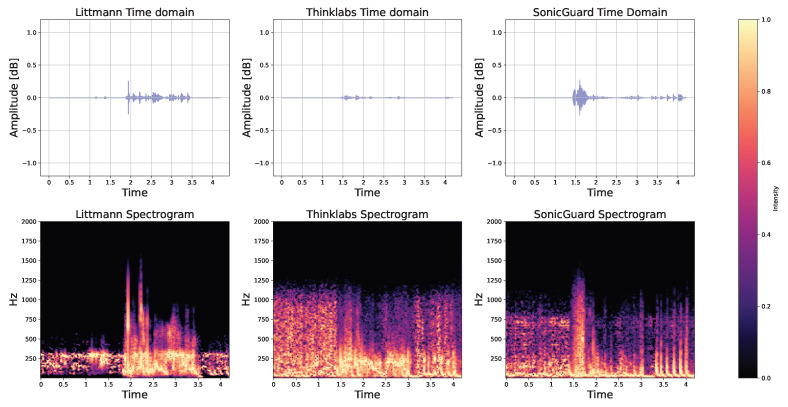
Bowel activity of Subject 001 recorded by Littmann, Thinklabs, and SonicGuard in time and frequency domain.

**Figure 10 sensors-24-01843-f010:**
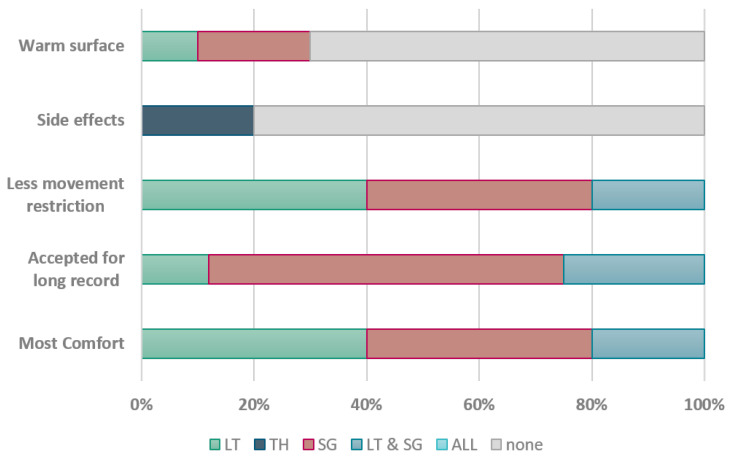
Subjects answer summary for the questionnaire starting from the first question on the bottom to the last question on the top, see Appendix D. (LT): Littmann, (TH): Thinklabs, (SG): SonicGuard (LT & SG): Littmann and SonicGuard, (ALL): all the three acoustic sensors, and (none): none of the acoustic sensors.

**Table 1 sensors-24-01843-t001:** Basic figures on SonicGuard and competing devices.

Dimension	Littmann	Thinklabs	SonicGuard
Weight [g]	232	50	10
Diameter [mm]	46	46	20
Height [mm]	25	28	8
Frequency range [Hz]	20–20,000	20–10,000	20–20,000
Battery life [hours]	8	4	24

**Table 2 sensors-24-01843-t002:** Power spectrum density of the bowel sound signal [$10^{-9}V^{2}/\mathrm{Hz}$] in the frequency range of interest [100–1000 Hz] recorded using Littmann (LT), Thinklabs (TH) and SonicGuard (SG) sensors under different conditions (normal, hypoactive, hyperactive). The first column defines the name of the active abdominal quadrant during the recording. The second column defines the location of the audio sensors during the recording. The names of the active quadrants during the recording are as follows: (ALL): refers to all the quadrants, (RUQ): right upper quadrant, (LUQ): left upper quadrant, (RLQ): right lower quadrant, (LLQ): left lower quadrant. The highest PSD value across each test subgroup is highlighted in bold.

Sound Source	Sensor Location	Normal			Hyperactive			Hypoactive		
		LT	TH	SG	LT	TH	SG	LT	TH	SG
ALL	Centre	5.65	0.56	**8.65**	**0.21**	0.04	0.17	0.19	0.28	**8.88**
RUQ	RUQ	6.48	30.2	**84.76**	0.18	0.05	**0.22**	0.18	0.05	**0.27**
LUQ	LUQ	1.12	0.96	**10.3**	**0.21**	0.51	0.11	0.18	0.05	**0.62**
RLQ	RLQ	18.12	32.24	**37.31**	0.14	0.005	**2.14**	0.18	0.05	**1.65**
LLQ	LLQ	**38.41**	18.19	35.3	13.85	4.54	**25.23**	0.17	0.05	**0.77**
RUQ	LUQ	0.18	0.005	**1.29**	0.18	0.05	**0.29**	0.22	0.05	**0.23**
RUQ	RLQ	0.22	0.005	**3.23**	0.19	0.05	**0.34**	0.19	0.05	**0.38**
RUQ	LLQ	0.18	0.005	**0.67**	0.18	0.05	**0.95**	0.18	0.05	**1.13**

**Table 3 sensors-24-01843-t003:** Power spectral density [$10^{-9}V^{2}/\mathrm{Hz}$] upon varying the locations of sound source and sensor location. (SG): SonicGuard sensor, (RUQ): right upper quadrant, (LUQ): left upper quadrant, (RLQ): right lower quadrant, (LLQ): left lower quadrant. The highest PSD value across each test subgroup is highlighted in bold.

Sound Source	Normal				Hyperactive				Hypoactive			
	SG RUQ	SG LUQ	SG RLQ	SG LLQ	SG RUQ	SG LUQ	SG RLQ	SG LLQ	SG RUQ	SG LUQ	SG RLQ	SG LLQ
RUQ	**69.26**	19.80	7.87	1.31	**68.07**	22.13	3.81	14.93	**97.13**	17.76	8.86	1.31
LUQ	2.67	**31.5**	0.65	23.35	1.99	**27.80**	9.89	8.01	1.22	**43.55**	1.159	2.53
RLQ	51.04	18.10	**155.4**	3.78	49.7	17.43	**80.27**	5.61	52.96	16.53	**104.84**	3.78
LLQ	49.26	22.79	94.74	**202.23**	49.27	18.67	64.14	**113.35**	50.19	21.01	68.29	**250.23**

**Table 4 sensors-24-01843-t004:** This table shows how the mean and standard deviation for the time domain parameters such as dynamic range, zero crossing, and fractal dimension for the three acoustic sensor Littmann (LT), Thinklabs (TH), and SonicGuard (SG) before and after applying a band-pass filter [20–1000 Hz]. The highest value across each features subgroup is highlighted in bold.

	Dynamic Range		Zero Crossing		Fractal Dimension	
	No Filter	With Filter	No Filter	With Filter	No Filter	With Filter
Littmann	**1.30 ± 0.65**	**1.99 ± 0.99**	229.5 ± 41.1	187.95 ± 38.28	1.07 ± 0.05	1.01 ± 0.004
Thinklabs	0.23 ± 0.15	0.32 ± 0.23	**1803.4 ± 528.3**	**202.66 ± 51.44**	**1.63 ± 0.23**	1.03 ± 0.01
SonicGuard	0.70 ± 0.35	0.5 8± 0.28	241.65 ± 79.5	143.38 ± 55.80	1.33 ± 0.12	**1.04 ± 0.01**

## Data Availability

The data is unavailable due to privacy and ethical restrictions.

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
