# Peer review of "SonicGuard Sensor—A Multichannel Acoustic Sensor for Long-Term Monitoring of Abdominal Sounds Examined through a Qualification Study"

_sensors, 2024, doi:10.3390/s24061843_

Round 1
Reviewer 1 Report
Comments and Suggestions for Authors
The submission paper describes the development and testing of the SonicGuard sensor, a multi-channel acoustic sensor designed for the long-term monitoring of abdominal sounds, particularly focusing on bowel sounds. The context outlines a series of qualification tests ranging from controlled experiments to real patient recordings, comparing the SonicGuard sensor's performance with commercial digital stethoscopes. The results indicate that SonicGuard provides comparable or superior performance, demonstrating its potential for enhancing machine learning-based analysis of bodily sounds for diagnostic purposes.
The following comments could be considered to improve the manuscript:
1) The Introduction section should be rewritten. The basic concepts could be shorten, and the related research and applications could be described.
2) It is suggested listing the specifications of each part of the sensor system, such as its frequency range, sensitivity, and data processing algorithms in Section 2.1.
3) The informed consent approval information should be added in Section 2.2.4 Test with Patients. The "Sensor performance" should revised as "Subject information".
4) The classification evaluation with only a support vector machine is not enough, because the SVM may sometimes lead to overfitting. It is suggested including some other benchmark classifiers in the performance evaluation.
Author Response
- The Introduction section should be rewritten. The basic concepts could be shortened, and related research and applications could be described.
Authors: The introduction has been adjusted, and a related research and applications section has been added.
- It is suggested to list the specifications of each part of the sensor system, such as its frequency range, sensitivity, and data processing algorithms in Section 2.1.
Authors: Some specifications of each part of the sensor system have been added in Table 1, such as the frequency range and battery life. Unfortunately, the sensitivity of Littmann and Thinklabs is not available in the sensor manuals. Additionally, all data processing algorithms for all sensors were disabled at the beginning of the study to ensure an equitable comparison.
- The informed consent approval information should be added in Section 2.2.4 Test with Patients. The "Sensor performance" should be revised as "Subject information".
Authors: Sensor performance section has been updated and revised as "Subject information" and now also includes information on the study protocol and informed consent.
- The classification evaluation with only a support vector machine is not enough because the SVM may sometimes lead to overfitting. It is suggested to include some other benchmark classifiers in the performance evaluation.
Authors: We thank the reviewer for bringing up this important point. The results of using different classifiers such as KNN and DTC have been added in supplementary material Appendix C. They show a similar ranking across the three sensors as data sources but are outperformed by the SVM classifier. For this reason, we decided to present only the SVM classifier in the main text.
Reviewer 2 Report
Comments and Suggestions for Authors
The authors have developed an interesting compact medical acoustic sensor. They have characterized the tool with promising results. But I have some comments for the authors before I can suggest any decision.
- The manuscript needs to be checked for English grammar errors and be revised accordingly. Also, some terms are not wrong, but they are better to be revised. For example, “Considered” is not needed in subsection 2-1. Or the term “real” is not needed in 2.2.4. Also, SNR is generally considered as “signal to noise ratio”, and not “self noise”.
- A more clear real picture of the proposed device is needed. This picture needs to show different parts of the proposed sensor clearly. Currently, a small picture of the casing only is shown.
- What Figures 4 and 5 shows? I cannot see agreement between the two devices in Figure 4. Figure legends are needed. Also, the texts are too small and hard to read. After reading the main text and the figure caption, I still have hard time understanding these two figures. Authors need to revise this section carefully.
- In the caption of Figure 4, the authors talk about the “yellow” box. But I could not find it.
- Figures 7-9 need scale bar.
Comments on the Quality of English Language
The text needs to be revised carefully. Please see the comments to the authors.
Author Response
- The manuscript needs to be checked for English grammar errors and be revised accordingly.
Authors: The entire manuscript has been checked for English grammar errors.
- Also, some terms are not wrong, but they are better to be revised. For example, “Considered” is not needed in subsection 2-1. Or the term “real” is not needed in 2.2.4. Also, SNR is generally considered as “signal to noise ratio,” and not “self-noise.”
Authors: We thank the reviewer for the careful reading of the manuscript and for the valuable suggestions. All mentioned terms have been fixed throughout the entire manuscript.
- A clearer real picture of the proposed device is needed. This picture needs to show different parts of the proposed sensor clearly. Currently, only a small picture of the casing is shown.
Authors: We again thank the reviewer for making us aware of this shortcoming. We included a picture of the proposed device along with corresponding description in Appendix A.
- What do Figures 4 and 5 show? I cannot see agreement between the two devices in Figure 4. Figure legends are needed. Also, the texts are too small and hard to read. After reading the main text and the figure caption, I still have a hard time understanding these two figures. Authors need to revise this section carefully.
Authors: A new picture has been added to Figure 4 to explain the experiment better, and the caption has been updated. The text size has been increased, and a legend has been added.
- In the caption of Figure 4, the authors talk about the “yellow” box. But I could not find it.
Authors: We thank the reviewer for the careful reading. The mistake has been fixed in Figure 4 caption.
- Figures 7-9 need a scale bar.
Authors: A scale bar has been added to Figures 7-9, and the text size has been increased.
Round 2
Reviewer 1 Report
Comments and Suggestions for Authors
The revised manuscript has been greatly improved.